

# Relationships between the concentration of particulate organic nitrogen and the inherent optical properties of seawater in oceanic surface waters

Alain Fumenia[1], Hubert Loisel[1], Rick A. Reynolds[2], and Dariusz Stramski[2]

[1]Laboratoire d'Océanologie et de Géosciences, Université du Littoral Côte d'Opale, Université Lille, CNRS, IRD, UMR 8187, LOG, Wimereux, France
[2] Marine Physical Laboratory, Scripps Institution of Oceanography, University of California San Diego, La Jolla, California
92093-0238, USA

*Correspondence to*: Alain Fumenia (alain.fumenia@univ-littoral.fr)

**Abstract.**

The concentration of particulate organic nitrogen (PON) in seawater plays a central role in ocean biogeochemistry. Limited availability of PON data obtained directly from in situ sampling methods hinders development of thorough understanding and

characterization of spatio-temporal variability of PON and associated source and sink processes within the global ocean. Measurements of seawater inherent optical properties (IOPs) that can be performed over extended temporal and spatial scales from various in situ and remote-sensing platforms represent a valuable approach to address this gap. We present the analysis of relationships between PON and particulate IOPs including the absorption coefficients of total particulate matter, $a_p(\lambda)$, phytoplankton, $a_{ph}(\lambda)$, and non-algal particles, $a_d(\lambda)$, as well as the particulate backscattering coefficient, $b_{bp}(\lambda)$. This analysis

is based on an extensive field dataset of concurrent measurements of PON and particulate IOPs in the near-surface oceanic waters and shows that reasonably strong relationships hold across a range of diverse oceanic and coastal marine environments. The coefficient $a_p(\lambda)$ and $a_{ph}(\lambda)$ show the best ability to serve as PON proxies over a broad range of PON from open ocean oligotrophic to coastal waters. The particulate backscattering coefficient can also provide a good proxy of PON in open ocean environments. The presented relationships demonstrate a promising means to assess PON from optical measurements

conducted from spaceborne and airborne remote-sensing platforms and in situ autonomous platforms. In support of this potential application, we provide the relationships between PON and spectral IOPs at light wavelengths consistent with those used by satellite ocean color sensors.



## 1 Introduction

Oceanic organic matter consists of dissolved (Dissolved Organic Matter, DOM) and particulate (Particulate Organic Matter, POM) constituents which span a wide range of sizes from the molecular scale to large particles suspended in water (Verdugo et al., 2004). POM, defined operationally as organic material captured on filters with nominal pore sizes ranging from 0.2 to 0.7 μm, includes large viruses, bacteria, phytoplankton, zooplankton as well as detrital material (Riley, 1971; Eppley et al., 1977; Eppley et al., 1983; Morel and Ahn, 1991; Stramski et al., 2004; Karhbush et al., 2020). The elemental composition of

the POM pool is made up of, among other constituents, particulate organic carbon, nitrogen, and phosphorus. Despite their important roles in ocean biogeochemistry, observations of mass concentrations of particulate organic carbon (POC), particulate organic nitrogen (PON), and particulate organic phosphorus (POP) obtained from direct measurement methods are relatively scarce, especially in terms of representing extended temporal and spatial scales of variability within the global ocean (Martiny et al., 2013).

To overcome these limitations, the seawater inherent optical properties (IOPs) such as the spectral particulate beam attenuation $c_p(\lambda)$, spectral particulate scattering $b_p(\lambda)$, spectral particulate backscattering $b_{bp}(\lambda)$, and spectral particulate absorption $a_p(\lambda)$ coefficients, measured in situ or estimated from spaceborne remote-sensing platforms using Ocean Color Radiometry (OCR), have been used as proxies of some ocean biogeochemical parameters, including POC and the mass concentration of suspended particulate matter, SPM. All these IOP coefficients are in units of m$^{-1}$, λ represents the wavelength

of light in vacuum in units of nm, the subscript "p" indicates that the IOP coefficients are associated with suspended particles in water, and the mass concentrations of particulate matter or its organic components are typically expressed in units of mg m$^{-3}$. To the first order, the variability in particulate IOPs is driven by total concentration of suspended particulate matter as well as composition and size distribution of suspended particles. Therefore, relationships between particulate IOPs representing the effects of all suspended particles and some measures of concentration of organic particles such as POC exhibit large variations

across diverse water bodies within the global ocean because of large changes in composition and size distribution of particulate matter. For example, large variations were demonstrated in the relationships between POC and $b_p(\lambda)$, $b_{bp}(\lambda)$, and $a_p(\lambda)$ across marine environments with highly variable proportions of organic and mineral particulate matter (Woźniak et al., 2010; Reynolds et al., 2016; Koestner et al., 2022; Stramski et al., 2023; Koestner et al., 2024). These studies indicated a need and proposed some approaches to account for variations in particulate composition when particulate IOPs are intended to be used

as proxies for POC, especially when a wide range of aquatic environments with highly variable characteristics of particulate assemblages is considered.

    Notwithstanding these challenges, many studies in the past have used in situ measurements conducted in different geographically restricted regions of the global ocean to demonstrate that the relationships between POC and particulate IOPs can be reasonably good under conditions which are regionally or environmentally constrained in terms of ocean bio-optical

properties. For example, such relationships were examined between POC and $c_p(\lambda)$ (e.g., Marra et al., 1995; Loisel and Morel, 1998; Bishop, 1999; Claustre et al., 1999; Stramska and Stramski, 2005; Gardner et al., 2006; Bishop and Wood, 2008; Cetinić



et al., 2012), between POC and $b_{bp}(\lambda)$ (e.g., Stramski et al., 1999; Stramski et al., 2008; Allison et al., 2010; Loisel et al., 2011; Cetinić et al., 2012; Kheireddine et al., 2020; Qiu et al., 2021; Barbieux et al., 2022), and between POC and $a_p(\lambda)$ (e.g., Woźniak et al., 2011; Rasse et al., 2017). While recognizing that single empirical relationships for estimating POC from particulate

IOPs are expected to work best if they are region-specific or formulated over a restricted range of marine bio-optical environments, it is also reasonable to assume that such relationships can be useful for applications across vast areas of open-ocean pelagic environments because the variations in particulate characteristics in these environments are expected to be constrained to a significant degree compared to variations observed across all diverse water bodies. For example, these relationships have been used to assess carbon community production from in situ $c_p(\lambda)$ or $b_{bp}(\lambda)$ measurements in the tropical

Pacific (Claustre et al., 1999), the South Pacific gyre (Claustre et al., 2008), and the Mediterranean Sea (Loisel et al., 2011; Barbieux et al., 2022). Vertical fluxes of POC have been described and quantified using POC vs. $b_{bp}$ relationships applied to in situ $b_{bp}(\lambda)$ measurements acquired from autonomous platforms during a sub-polar North Atlantic spring bloom (Briggs et al., 2011) and in the Red Sea (Kheiredinne et al., 2020). Based on the POC vs. $b_{bp}(\lambda)$ relationships the ocean color algorithms have also been developed (e.g., Stramski et al., 1999; Loisel et al., 2001; Loisel et al., 2002; Stramska and Stramski, 2005;

Allison et al., 2010; Duforet et al., 2010) to enhance the capabilities and complement other algorithms that have been used for estimating POC in surface waters of the global ocean from satellite ocean color observations (Stramski et al., 2008; Stramski et al. 2022).

Considering the interest of developing a capability to estimate PON from optical measurements along with the existing algorithms that allow the estimation of POC from optical measurements, it is relevant to comment on the canonical Redfield

ratio which describes a consistent atomic ratio of carbon, C, nitrogen, N, and phosphorus, P, in marine plankton, namely C:N:P of 106:16:1 (Redfield et al., 1934; 1963). This ratio could potentially serve as a means to estimate PON from POC. However, it is well recognized that the C:N:P ratios for natural particulate organic matter can vary considerably in the ocean, and thus deviate from the canonical Redfield ratio (Copin-Montegut and Copin-Montegut, 1983; Diaz et al, 2001; Körtzinger et al., 2001; Geider and Laroche, 2002; Weber and Deutsch, 2010; White et al., 2006). For example, strong latitudinal patterns in

these elemental ratios of marine plankton and particulate organic matter that includes also non-living particles, such as detritus generated from the decay of phytoplankton cells and zooplankton grazing activity, has been documented (Martiny et al., 2013). Therefore, the subject of estimating PON from optical measurements requires separate dedicated studies and this paper is a contribution to this line of research.

Recently, a reasonably good relationship between PON and $b_{bp}$ was demonstrated based on field measurements made in

oligotrophic waters of the western tropical South Pacific (Fumenia et al., 2020). Furthermore, this study used the $b_{bp}$ measurements from Biogeochemical-Argo (BGC-Argo) floats to quantify new production of phytoplankton biomass likely related to intense biological nitrogen ($N_2$) fixation in this tropical oceanic environment. It is worth noting that at the scale of the global ocean, the biological $N_2$ fixation is a major source of new nitrogen in the euphotic layer, followed by atmospheric and terrestrial deposition (Dugdale, 1961; Karl et al., 2002; Capone et al., 2005). Also, different models utilizing a combination

of in situ PON measurements and satellite ocean color observations, including satellite-derived ocean remote-sensing



reflectance $R_{rs}(\lambda)$ as well as satellite-derived IOPs such as $b_{bp}(\lambda)$ and the total absorption coefficient of seawater $a(\lambda)$, have been developed for applications at the global oceanic scale (Wang et al., 2022). Although the study of Wang et al. (2022) suggests that PON can be estimated from satellite ocean color products, the PON vs. IOPs relationships have not yet been investigated using field measurements collected over a broad range of oceanic and coastal marine environments.

The main objective of the present study is to examine the relationships between the PON and particulate IOPs, including $b_{bp}(\lambda)$, $a_p(\lambda)$, $a_{ph}(\lambda)$, and $a_d(\lambda)$, from in situ near-surface measurements collected over a broad range of marine bio-optical environments. For this purpose, we assembled datasets of concurrent PON and IOP measurements from the open-ocean pelagic environments, Arctic seas, and coastal waters around Europe. The relationships between PON and spectral IOPs are presented and discussed in terms of variability and its sources as observed in the examined relationships across the different marine

environments. This analysis provides insights into the potential applicability of different particulate IOPs to serve as proxies for PON.

## 2 Materials and methods

### 2.1 Geographic locations of in situ measurements

A dataset of in situ biogeochemical and optical measurements was assembled from multiple field experiments performed in

various open ocean and coastal regions covering a broad range of PON, POC, and particulate IOPs collected at depths between the sea surface and 10 m (Fig. 1; Table 1). The whole dataset (referred to as WD) consists of three subsets of data collected in different regions which generally represent different marine bio-optical environments.

The first subset of data (referred to as OOD for the open-ocean dataset) includes measurements made during three cruises conducted in open ocean waters in the Pacific and Atlantic Oceans. The BIOSOPE (Biogeochemistry and Optics South Pacific

Experiment) cruise took place from October to December 2004 in the eastern South Pacific Ocean along an east-to-west transect from the Marquesas Islands to the coast of Chile (Claustre et al., 2008; Stramski et al., 2008). The KM12-10 cruise was carried out in June 2012 in tropical waters off the Hawaiian Islands (Johnsen et al., 2014; Reynolds and Stramski, 2021). The ANTXXVI/4 cruise was conducted in April and May 2010 along a south-to-north transect in the Atlantic Ocean between Chile and Germany (Uitz et al., 2015).

The second subset of data (referred to as AOD for the Arctic Ocean dataset) includes measurements collected in the western Arctic seas, specifically in the Chukchi Sea and western Beaufort Sea during three cruises, HLY1001 in June-July 2010, HLY1101 in June-July 2011, and MR17-05C in August-September 2017 (Arrigo, 2015; Reynolds and Stramski, 2019; Shiozaki et al., 2019). The data collected in these high latitude environments are characterized by the presence of specific phytoplankton communities and a relatively high contribution of dissolved organic matter (CDOM) and nonalgal particulate

matter to IOPs of seawater (Reynolds and Stramski, 2019).

The third subset of data (referred to as CWD for the coastal-water dataset) consists of data collected as part of the COASTlOOC (Coastal Surveillance Through Observation of Ocean Color) research project which involved numerous





experiments in various coastal waters around Europe in 1997 and 1998 (Massicotte et al., 2023). This dataset represents the bio-optical variability encountered across diverse coastal waters including shelf and relatively shallow environments in the

Baltic Sea, North Sea, Wadden Sea, English Channel, and Adriatic Sea as well as waters affected by many river plumes around Europe (Babin et al., 2003a; 2003b). A small fraction of CWD (< 3 % of COASTlOOC data) includes measurements collected in open ocean waters in the Atlantic Ocean between the Bay of Biscay and the Canary Islands and off the shelf in the Mediterranean Sea, where the bio-optical variability is expected to be driven primarily by phytoplankton and associated material.

The total number of concurrent POC and PON measurements, $N_{POM}$, in the whole dataset WD is 432 (Table 1). The contributions of OOD, AOD, and CWD to this total number are 18.8 %, 25.2 %, and 57.4 %, respectively. These measurements of PON and POC are used to discuss the variability in the POC/PON ratio in Sect. 3.1. The number of concurrent measurements of PON and IOPs which are used to examine the relationships between these variables is smaller than $N_{POM}$. Specifically, the presented relationship between PON and the backscattering coefficient $b_{bp}$ is based on 284 measurements in the whole dataset

WD (Sect. 3.2.1) and the relationships between PON and the absorption coefficients, $a_p$, $a_{ph}$, and $a_d$, are based on 392 measurements (Sect. 3.2.2).

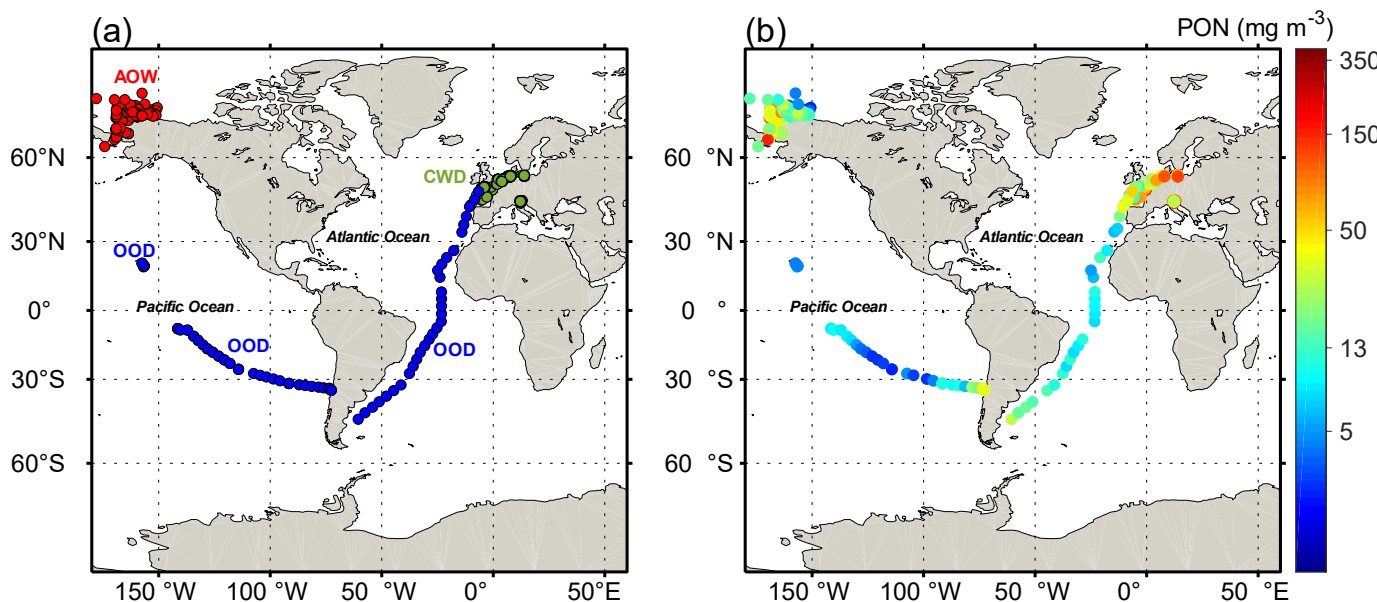

**Figure 1. (a) Geographical locations of oceanographic stations shown as color coded symbols according to the three subsets of data from different oceanic regions. OOD, AOD, and CWD refer to the datasets for the open-ocean Pacific and Atlantic waters, western**

**Arctic seas, and European coastal waters, respectively. (b) Near-surface PON measured at all stations in the whole dataset (WD).**



**Table 1. The values of the minimum-to-maximum range (in brackets) and median (in parentheses) for the whole dataset (WD) and different subsets of data used in this study. The data subsets are OOD (open-ocean dataset), AOD (Arctic Ocean dataset), and CWD (coastal-water dataset). The results are also shown for the three oceanic regions which are part of OOD. $N_{\mathrm{POM}}$ is the number of PON or POC measurements. $N_{\mathrm{bbp}}$ is the number of concurrent PON and $b_{\mathrm{bp}}(555)$ measurements. $N_{\mathrm{ap}}$ is the number of concurrent PON and $a_{\mathrm{p}}(510)$, $a_{\mathrm{ph}}(510)$, and $a_{\mathrm{d}}(442)$ measurements.**

| North Pacific | Atlantic | CWD (open waters) | OOD | AOD | CWD |
|---|---|---|---|---|---|
| [4.3–7.7] (5.4) | [5.2–55.3] (10.8) | [6.0–19.0] (10.0) | [2.4–68.8] (9.05) | [2.5–151.0] (22.5) | [6.0–340.0] (55.0) |
| [23.0–44.8] (31.1) | [23.8–257.5] (52.5) | [40.0–126.0] (46.5) | [11.9–351.7] (43.9) | [15.6–1022.1] (128.3) | [40.0–2470.0] (270.0) |
| [6.3–6.9] (6.6) | [4.8–6.5] (5.6) | [2.8–9.7] (7.9) | [2.8–9.7] (5.8) | [4.9–20.1] (6.9) | [2.2–16.4] (7.3) |
| [4.1×10⁻⁴–1.1×10⁻³] (4.5×10⁻⁴) | [4.5×10⁻⁴–2.7×10⁻³] (7.8×10⁻⁴) | no data | [3.3×10⁻⁴–3.7×10⁻³] (8.1×10⁻⁴) | [2.9×10⁻⁴–0.2] (2.6×10⁻³) | [3.8×10⁻³–1.3] (2.2×10⁻²) |
| [2.2×10⁻³–4.8×10⁻³] (3.4×10⁻³) | [3.7×10⁻³–5.1×10⁻²] (5.4×10⁻³) | [2.3×10⁻³–2.5×10⁻²] (8.8×10⁻³) | [6.4×10⁻⁴–5.1×10⁻²] (4.5×10⁻³) | [1.9×10⁻³–0.5] (2.0×10⁻²) | [2.3×10⁻³–0.6] (8.4×10⁻²) |
| [1.6×10⁻³–2.7×10⁻³] (2.1×10⁻³) | [1.1×10⁻³–4.3×10⁻²] (3.4×10⁻³) | [1.6×10⁻³–1.9×10⁻²] (7.6×10⁻³) | [3.5×10⁻⁴–3.5×10⁻⁴] (3.0×10⁻³) | [6.4×10⁻⁴–0.1] (8.1×10⁻³) | [1.6×10⁻³–0.4] (5.7×10⁻²) |
| [1.0×10⁻³–3.9×10⁻³] (2.2×10⁻³) | [1.8×10⁻³–1.3×10⁻²] (4.5×10⁻³) | [1.9×10⁻³–1.4×10⁻²] (2.9×10⁻³) | [3.8×10⁻⁴–1.3×10⁻²] (2.4×10⁻³) | [1.6×10⁻³–0.7] (1.7×10⁻²) | [1.9×10⁻³–0.9] (6.7×10⁻²) |
| [0.49–0.69] (0.56) | [0.39–0.69] (0.54) | [0.25–0.68] (0.49) | [0.25–0.69] (0.53) | [2.40×10⁻²–0.62] (0.36) | [1.62×10⁻²–0.68] (0.17) |
| [0.53–0.77] (0.65) | [0.23–0.92] (0.65) | [0.71–0.89] (0.81) | [0.23–0.92] (0.73) | [0.01–0.86] (0.51) | [0.23–0.88] (0.64) |
| 8 | 25 | 6 | 81 | 109 | 248 |
| 8 | 25 | no data | 74 | 106 | 104 |
| 8 | 25 | 5 | 71 | 92 | 229 |




| Datasets | WD | South Pacific |
|---|---|---|
| PON (mg m$^{-3}$) | [2.4–340.0] (34.2) | [2.4–68.8] (9.1) |
| POC (mg m$^{-3}$) | [11.9–2470.0] (180.0) | [11.9–351.7] (43.2) |
| POC/PON (g/g) | [2.0–17.2] (5.8) | [4.0–7.0] (5.7) |
| $b_{bp}$(555) (m$^{-1}$) | [2.9×10$^{-4}$–0.3] (3.5×10$^{-2}$) | [3.3×10$^{-4}$–3.7×10$^{-3}$] (9.6×10$^{-4}$) |
| $a_p$(510) (m$^{-1}$) | [6.4×10$^{-4}$–0.6] (4.7×10$^{-2}$) | [6.4×10$^{-4}$–4.7×10$^{-2}$] (3.1×10$^{-3}$) |
| $a_{ph}$(510) (m$^{-1}$) | [3.5×10$^{-4}$–0.4] (2.7×10$^{-2}$) | [3.5×10$^{-4}$–4.1×10$^{-2}$] (2.5×10$^{-3}$) |
| $a_d$(442) (m$^{-1}$) | [3.8×10$^{-4}$–0.9] (3.5×10$^{-2}$) | [3.8×10$^{-4}$–1.2×10$^{-2}$] (1.3×10$^{-3}$) |
| POC/SPM (g/g) | [1.62×10$^{-2}$–0.69] (0.24) | [0.25–0.69] (0.50) |
| $a_{ph}$(510)/ $a_p$(510) | [0.01–0.92] (0.63) | [0.40–0.92] (0.82) |
| $N_{POM}$ | 438 | 42 |
| $N_{bbp}$ | 284 | 41 |
| $N_{ap}$ | 392 | 33 |

## 2.2 Measurement Methods

The measurement and data processing protocols are described in detail in the references cited in Sect. 2.1. Here we provide a brief summary. Samples for POC and PON determinations were collected by filtration of seawater through pre-combusted 25-mm Whatman GF/F filters. After filtration, the samples were transferred into glass vials, dried at 55°C, and stored until post-cruise analysis. The mass of particulate organic carbon and nitrogen on the sample filters was determined by high temperature combustion via standard CHN analysis following the JGOFS (Joint Global Ocean Flux Study) protocols (Knap et al., 1996). The samples were acid-treated prior to the CHN analysis to remove inorganic carbon. The mass concentration of suspended particulate matter (SPM) was determined gravimetrically by measuring the dry mass of particles collected on GF/F filters. The filters were pre-rinsed, pre-combusted, and pre-weighed using a protocol described in Van der Linde (1998). More details on the methodology of CHN analysis and SPM determinations are provided in Babin et al. (2003a), Reynolds et al. (2016), and Stramski et al. (2008).

The spectral particulate absorption coefficient, $a_p(\lambda)$, was determined on samples collected on 25-mm GF/F filters using a spectrophotometric filter-pad method. The $a_p(\lambda)$ spectra included in OOD and AOD were measured mostly with a Perkin-Elmer Lambda18 spectrophotometer equipped with a 15-cm integrating sphere using the inside integrating sphere (IS) configuration of measurement which is considered to provide the best accuracy of measurements with a filter-pad method (Stramski et al., 2015; Roesler et al., 2018). The exception is the absorption data from BIOSOPE cruise which were measured with a Perkin-Elmer Lambda 19 equipped with a 6-cm integrating sphere using a transmittance (T) configuration of measurement (Bricaud et al., 2010). The nonalgal particulate absorption coefficient, $a_d(\lambda)$, was also determined from the spectrophotometric filter-pad method after extraction of pigments (associated primarily with phytoplankton) in methanol



(Kishino et al., 1985). All absorption data in OOD and AOD were acquired between 300 and 800 nm with a 1 nm step. The

$a_p(\lambda)$ spectra included in CWD were determined from the transmittance-reflectance (T-R) configuration of filter-pad method

in the spectral range 380−750 nm at 1 nm intervals (Babin et al., 2003b). In this dataset, $a_d(\lambda)$ was determined by pigment

bleaching with sodium hypochlorite (Ferrari and Tassan, 1999). For all absorption samples considered in this study, the spectral

phytoplankton absorption coefficient, $a_{ph}(\lambda)$, was obtained by subtracting the measured $a_d(\lambda)$ from measured $a_p(\lambda)$. More details

on the absorption measurement methodology used in our dataset are provided in Babin et al. (2003b), Bricaud et al. (2010),

Uitz et al. (2015), and Reynolds and Stramski (2019).

The spectral backscattering coefficient $b_b(\lambda)$, which is the sum of particulate $b_{bp}(\lambda)$ and pure seawater $b_{bw}(\lambda)$ contributions,

was calculated from the scattering measurements at a specified backscattering angle (around 140°). After subtraction of $b_{bw}(\lambda)$

from $b_b(\lambda)$, the result was converted to $b_{bp}(\lambda)$ assuming a coefficient of proportionality between $b_{bp}(\lambda)$ and scattering at 140°.

The backscattering measurements for the datasets OOD and AOD were all performed with HydroScat-6 (HOBI Labs, Inc.)

instruments providing six wavelengths (420, 442, 470, 510, 555, 589 nm) during the BIOSOPE cruise and eleven wavelengths

(394, 420, 442, 510, 532, 550, 589, 620, 640, 671, 730, 852 nm) during the HLY1001, HLY1101, MR17-05C, ANT26, and

KM12-10 cruises. A more detailed description of the procedure to estimate $b_{bp}(\lambda)$ from HydroScat-6 measurements is provided

in Stramski et al. (2008) and Reynolds et al. (2016).

No in situ measurements of $b_{bp}(\lambda)$ were performed during the COASTlOOC experiments. However, in situ measurements

of downwelling, $E_d(z, \lambda)$, and upwelling, $E_u(z, \lambda)$, irradiances were conducted within the surface ocean layer at each station

(where $z$ is depth). From these vertical profiles, the irradiance reflectance just beneath the sea surface, $R(0^-, \lambda) = E_u(0^-, \lambda)/E_d(0^-, \lambda)$,

and the average attenuation coefficient for downwelling irradiance, $<K_d(\lambda)>_1 = 1/z_1$, between the surface and the first

attenuation depth was calculated ($0^-$ indicates the depth just beneath the sea surface, and $z_1$ is the first attenuation depth at

which the downwelling irradiance is reduced to about 36.8 % of its surface value). For the dataset CWD, $b_{bp}(\lambda)$ was then

estimated from the LS inverse optical model which uses $R(0^-, \lambda)$, $<K_d(\lambda)>_1$, and the sun angle as input parameters (Loisel and

Stramski, 2000). As $b_{bp}(\lambda)$ is driven largely by the concentration of suspended particulate matter, SPM, the reliability of LS-

derived $b_{bp}(\lambda)$ was assessed through comparison with $b_{bp}(555)$ obtained from a previously developed empirical relationship

(Neukermans et al., 2012) between $b_{bp}$ and SPM (Fig. 2a). The same inter-comparison exercise was performed using the

empirical relationship developed by Stramski et al. (2023) between SPM and $b_{bp}(555)$ (Fig. 2b). This comparative analysis

supports the use of LS-derived $b_{bp}(\lambda)$ for the COASTlOOC experiments considered in this study (Fig. 2). We note that similar

support was obtained with the use of the LS2 model (Loisel et al., 2018) instead of the LS model (not shown), where the main

difference is that the application of LS2 model requires the input of remote-sensing reflectance, $R_{rs}(\lambda)$, rather than $R(0^-, \lambda)$.

The measurements of PON, POC, and IOPs are subject to errors which are not amenable to straightforward quantification,

especially on a sample-by-sample basis. Multiple factors related to measurement methodology, instrumentation, environmental

conditions, and no knowledge of true values make it challenging to determine the errors. It has been common to use a series

of replicate observations for evaluating one of the components of measurement uncertainty. The precision of POC and PON

measurements performed during the BIOSOPE cruise has been determined from the analysis of duplicate and triplicate





samples. The coefficient of variation (CV) was, on average, about 8.7% and 7.7% for POC and PON, respectively (Stramski et al., 2008). For different cruises comprising the AOD dataset the median coefficient of variation for replicate samples of

POC varied between about 2% and 5% (Stramski et al., 2023) and a similar range of 3% to 4% was observed for PON. For the COASTlOOC experiment, the CV values were 3.7 % and 6.8 % for POC and PON, respectively (Ferrari et al., 2003). Thus, the precision of POC and PON measurements is expected to remain typically below 10%. Regarding the particulate absorption measurements, the lowest uncertainties below 10% with a high precision typically of a few percent are expected for the IS configuration of the filter pad method and the highest uncertainties are expected for the T configuration of this method

(Stramski et al., 2015; Roesler et al., 2018). While most absorption data in the OOD and AOD were obtained with the IS method, the T method was used during the BIOSOPE cruise for which the uncertainty of $a_p(\lambda)$ was estimated at about 15% while the precision based on replicate samples at a few percent in the visible spectral range (Bricaud et al., 2010). For the T-R method used during the the COASTlOOC experiment, the uncertainties are expected to be in between those for the IS and T methods (Stramski et al., 2015). Earlier analysis of situ determinations of backscattering coefficient indicated that the

uncertainty estimates commonly fall in the 10-20% range (e.g., Berthon et al. 2007; Doxaran et al. 2016) but can be reduced to a few percent under certain circumstances (Sullivan et al. 2013).

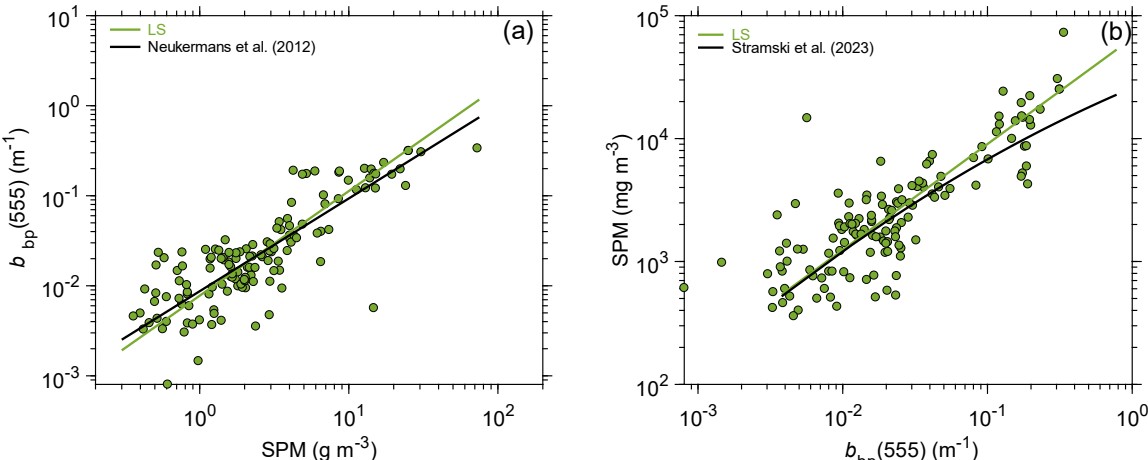

**Figure 2. (a) Scatter plot of $b_{bp}(555)$ as a function of SPM for the CWD dataset where $b_{bp}(555)$ is estimated from the LS model (green circles). The green line refers to the Model-II best linear fit using the log-transformed variables. For comparison, the $b_{bp}(555)$ vs.**
**SPM relationship of Neukermans et al. (2012) originally developed at 660 nm and recalculated for 555 nm assuming that $b_{bp}(\lambda)$ has a mean spectral dependency of $\lambda^{-0.5}$ (Babin et al., 2003a) is also shown (black line). (b) Same as (a) but for SPM as a function of $b_{bp}(555)$ for the CWD dataset. For comparison, the SPM vs. $b_{bp}(555)$ relationship of Stramski et al. (2023) is also shown (black line).**

## 2.3 Description of Dataset

The biogeochemical and optical properties exhibit a large variability in our whole dataset (WD) which is associated with a range of diverse marine bio-optical and trophic conditions in this dataset (Fig. 3, 4 and Table 1). Overall, PON (Fig. 3a) and



POC (Fig. 3b) range over 2 orders of magnitude from 2.4 to 340.0 mg m$^{-3}$ and 11.9 to 2470.0 mg m$^{-3}$, respectively. Minimum concentrations of these particulate organic pools are found in the subset of open-ocean data (OOD) with the smallest values of PON and POC observed in the oligotrophic waters of subtropical gyres in the Pacific and Atlantic Oceans (Table 1). The OOD

is also characterized by the highest median value of POC/SPM = 0.53 (Table 1) which is indicative of highly organic-dominated particulate assemblages. Maximum values of PON and POC are found in the COASTlOOC subset of data (CWD) which also has the lowest median value of POC/SPM = 0.17 (Table 1). This indicates that the coastal waters in the CWD generally have significantly smaller contribution of organic particles to SPM compared with the two other subsets of data (OOD and AOD). The median values of PON and POC in the European coastal environments are about 5 to 10 times higher

than those in the Pacific and Atlantic Oceans and twice as high as in the western Arctic seas (Table 1). Generally, the median values of biogeochemical variables in the Arctic dataset are between those for the open-ocean and coastal-water datasets. The range of variation in PON and POC is largest (about 60-fold) in the CWD and AOD whereas OOD has about 30-fold range of variation. The median value of POC/PON is 5.8 g/g(gram/gram) (≈ 6.8 mol/mol) for the whole dataset which is very similar to the canonical molar Redfield ratio of 6.6 mol/mol. However, the POC/PON ratio exhibits a large range of variability

(2.0−17.2 g/g) in our dataset (Fig. 3c, more details in Sect. 3.1).





**Figure 3.** Frequency distribution of near-surface values of (a) POC, (b) PON (c) POC/PON (g/g, i.e., gram/gram basis), and (d) POC/SPM (g/g) for the whole dataset (WD) used in this study. The black solid vertical lines correspond to the median values. The red vertical lines in panel (d) refers to the threshold values of POC/SPM of 0.12 and 0.28 which delimit mineral-dominated, mixed, and organic-dominated particulate assemblages (Stramski et al., 2023). The minimum-to-maximum range (Min-Max), median (Med), and coefficient of variation (CV) are also indicated. $N$ is the number of data.



The POC/SPM ratio expressed on a g/g basis (Fig. 3d) can be used as a proxy for characterizing the contributions of
organic versus inorganic particles to SPM. Some threshold values have been proposed to delimit the organic-dominated, mixed,
and mineral-dominated particulate assemblage in previous studies (Woźniak et al., 2010; Lubac and Loisel, 2007; Loisel et
al., 2023; Stramski et al., 2023). The threshold values established in different studies are similar; for example, Stramski et al.
(2023) proposed POC/SPM = 0.12 as a boundary between the mineral-dominated and mixed particulate assemblages and
POC/SPM = 0.28 as a boundary between the mixed and organic-dominated assemblages. Using these threshold values, we
determined that 43.9 % of our whole dataset is associated with organic-dominated particulate assemblages, 27.4 % with
mineral-dominated assemblages, and 28.7 % with mixed assemblages.

Similar to the biogeochemical parameters, the particulate IOPs exhibit a large range of variability which is illustrated in
Fig. 4 for the optical coefficients at selected light wavelengths. As expected, the IOP values are maximum in turbid coastal
waters included in the CWD dataset and minimum in the subtropical gyres included in the OOD (Table 1). The median value
of $b_{bp}(555)$ in the European coastal environments (CWD) is one order of magnitude higher than in the western Arctic seas
(AOD) and two orders of magnitude higher than in the open-ocean waters of the Pacific and Atlantic Oceans (OOD). The
median values of $a_p(510)$, $a_{ph}(510)$, $a_d(442)$ in the CWD are similar to those in the AOD but one order of magnitude higher
than those in the OOD (Table 1). The $a_{ph}(510)/a_p(510)$ ratio, which quantifies the proportion of phytoplankton absorption to
the total particulate absorption at light wavelength of 510 nm, varies by a factor of 83 within the whole dataset (see WD in
Table 1). Very high variability in this parameter is observed in the Arctic dataset (factor of 78), whereas the variation in the
open-ocean and coastal-water datasets is much smaller (about 4-fold).







**Figure 4. Frequency distribution of the near-surface values of (a) $b_{bp}(555)$, (b) $a_p(510)$, (c) $a_{ph}(510)$, and (d) $a_d(442)$ for the whole dataset (WD) used in this study. The black solid vertical lines correspond to the median values. The minimum-to-maximum range (Min-Max), median (Med), and coefficient of variation (CV) are indicated. $N$ is the number of data.**



## 2.4 Statistical Indicators

Model-I linear regression can be considered as a valid approach when the primary goal of analysis is to fit a predictive model to a dataset of the response ($y$) and explanatory ($x$) variables, i.e., to reduce variance in prediction of $y$ from $x$ (Legendre and Michaud, 1999; Sokal and Rohlf, 1995). In this study, the analysis of PON (response variable) versus IOPs (explanatory variables) is aimed at establishing the predictive relationships. Another alternative of linear regression analysis is Model-II which typically serves to quantify the strength of the linear relationship between the examined variables but can also be an adequate option for predictive purposes, especially when both variables are subject to error and the error in data of $x$ is not significantly smaller than the error in data of $y$ (McArdle, 1988). In our study, the uncertainties in the explanatory variables (IOPs) are not necessarily much smaller than in PON (see Section 2.2), so we tested both the Model-I and Model-II regressions. Specifically, we evaluated: (i) the ordinary least squares Model-I linear regression, (ii) the robust least squares Model-I linear regression, and (iii) the Model-II linear regression using the major axis method (Kermack and Haldane, 1950; York, 1966). These regression models were applied to the $\log_{10}$-transformed PON and IOP data. It should be noted that the Model-II linear regression using the major axis method is appropriate when both variables are expressed in the same physical units or are dimensionless (e.g., log-transformed variables) (Legendre and Legendre, 2012).

This analysis was made for the whole dataset, WD, and the three data subsets, OOD, AOD, and CWD. From this analysis we obtained the best-fit equations in the form of power function and the coefficient of determination ($R^2$) between the $\log_{10}$-transformed variables. The general formula of the power function is:

$$PON = A\ IOP(\lambda)^B \tag{1}$$

where $IOP(\lambda)$ represents one of the spectral particulate IOPs, $A$ and $B$ are the best-fit coefficients, and PON and $IOP(\lambda)$ variables are expressed in units of mg m$^{-3}$ and m$^{-1}$, respectively. For most examined cases, the general pattern of data points of PON vs. IOP is consistent with a power function; however, there is an exception for the relationship PON vs. $b_{bp}(\lambda)$ for the whole dataset WD. In this case, we also used a third-degree polynomial function that provided a better fit to data than the power function.

To compare the three methods of regression analysis we examined the goodness-of-fit of each regression equation (i.e., each PON algorithm utilizing a given particulate IOP as input to the algorithm) through the analysis of algorithm-derived PON vs. measured PON using the algorithm development dataset. This evaluation involved the use of Model-II linear regression based on the major axis method as applied to data of algorithm-derived vs. measured PON as well as the calculation of several statistical metrics that quantify differences between the algorithm-derived and measured values of PON (Table 2). These statistics include the slope ($S$) and the intercept ($I$) obtained from Model-II linear regression applied to $\log_{10}$-transformed variables of algorithm-derived vs. measured PON. These parameters are useful to reveal the potential presence of bias across the dynamic range of PON. The Median Bias ($MdB$) and the Median Ratio ($MdR$) quantify the aggregate systematic deviations



between the (non-transformed) algorithm-derived and measured values of PON for the investigated dataset. The Median
Absolute Percentage Difference (*MdAPD*) and the Root Mean Square Deviation (*RMSD*) characterize random deviations
between the algorithm-derived and measured PON. We also use the Median Symmetric Accuracy (*MdSA*) which can be
interpreted similarly to *MdAPD* as a median percentage difference but, unlike *MdAPD*, *MdSA* does not penalize over- and
under-prediction differently (Morley et al., 2018).

The comparative analysis of algorithm-derived PON vs. measured PON indicated that some statistics are better for the
predictive regression formulas obtained from the Model-II regression compared with the predictive formulas obtained from
Model-I regression analysis. Specifically, this improvement was observed for the slope (*S*) and the intercept (*I*) of log-
transformed algorithm-derived PON vs. measured PON. Other statistics did not reveal any advantage of Model-II over Model-
I regression (or vice versa) for establishing the empirical algorithms for PON vs. IOPs. As a result of this analysis, in the
remainder of this study all presented PON vs. IOP algorithms are based on Model-II regression using the major axis method
as applied to the $\log_{10}$-transformed variables.

**Table 2. Statistical metrics used in the evaluation of the goodness-of-fit of algorithmic formulas.**

| Symbol | Description |
|---|---|
| $y_i$, $x_i$ (mg m$^{-3}$) | Algorithm-derived PON ($y_i$) and measured PON ($x_i$) for sample $i$ of $N$ |
| $N$ | Number of samples (data) |
| $R$ | Pearson's product moment correlation coefficient between $\log_{10}$-transformed variables used in Model-II linear regression |
| $S$ and $I$ | Slope and intercept obtained from Model-II linear regression |
| $MdB$ (mg m$^{-3}$) | Median Bias; median value of $(y_i - x_i)$ |
| $MdR$ | Median Ratio of $(y_i/x_i)$ |
| $MdAPD$ (%) | Median Absolute Percentage Difference; median value of $100 \times [|(y_i - x_i)/x_i|]$ |
| $MdSA$ (%) | Median Symmetric Accuracy; $100 \times [10^{\text{median}[|\log(y_i/x_i)|]} - 1]$ |
| $RMSD$ (mg m$^{-3}$) | Root Mean Square Deviation; $[(1/N) \sum^N (y_i - x_i)^2]^{0.5}$ |

For the final algorithms based on Model-II regression analysis, we further evaluated the relationships between the
algorithm-derived and measured PON using the radar charts (Tran et al., 2019). For this purpose, *MdB, MdAPD, MdSA, RMSD,*
*S*, and *R* were normalized as follows:

$$MdB_{norm}(j) = \frac{|MdB(j)|}{\max\left(|MdB(j)|,\ j = 1, k\right)}$$



$$MdAPD_{norm}(j) = \frac{MdAPD(j)}{\max\left(MdAPD(j),\ j=1,k\right)}$$


$$MdSA_{norm}(j) = \frac{MdSA(j)}{\max\left(MdSA(j),\ j=1,k\right)}$$

$$RMSD_{norm}(j) = \frac{RMSD(j)}{\max\left(RMSD(j),\ j=1,k\right)}$$


$$S_{norm}(j) = \frac{|1-S(j)|}{\max\left(|1-S(j)|,\ j=1,k\right)}$$

$$R_{norm}(j) = \frac{\min\left(R(j), j=1,k\right)}{R(j)}$$

where $j$ represents each individual PON algorithm based on a given IOP and $k$ is the number of tested algorithms. In addition, to facilitate the comparison between the goodness-of-fit of PON algorithms based on different IOPs, the area associated with the polygons linking the normalized statistical indicators were computed as:

$$\begin{aligned}
Area = \frac{1}{2} \times \frac{\pi}{6} \times [ &RMSD_{norm}(j) \times MdAPD_{norm}(j) + MdAPD_{norm}(j) \times MdSA_{norm}(j) \\
&+ MdSA_{norm}(j) \times MdB_{norm}(j) + MdB_{norm}(j) \times S_{norm}(j) + S_{norm}(j) \times R_{norm}(j) \\
&+ R_{norm}(j) \times RMSD_{norm}(j)]
\end{aligned}$$


## 3 Results and Discussion

### 3.1 POC vs. PON Relationship

The carbon-to-nitrogen ratio of the organic particulate matter, POC/PON, in our dataset and deviations in these data from the
canonical Redfield ratio of 106/16 mol/mol (approximately 6.6) are depicted in Fig. 5. We note that in this illustration we present POC and PON in micromolar concentration units because such units were used in the original work on the Redfield ratio (Redfield et al., 1934; 1963). POC and PON are generally well correlated with $R = 0.85$ for the whole dataset (WD) but the linear regression fitted to the POC vs. PON data deviates slightly from the relationship corresponding to the canonical Redfield ratio (Fig. 5a). For WD the slope $S$ of the best-fit linear function is $6.18 \pm 0.18$ which is lower than the Redfield ratio
value.





When considering the three subsets of data, the open-ocean data (OOD) exhibit the strongest correlation between POC and PON ($R$ = 0.99, Fig. 5a) and the lowest variability in POC/PON (coefficient of variation CV = 15.5 %, Fig. 5b). The median value of POC/PON for open-ocean data is 5.8 (Table 1, Fig. 5b) which is significantly lower than the Redfield ratio. In striking contrast to OOD, the coastal-water dataset (CWD) has the lowest correlation between POC and PON ($R$ = 0.75,

Fig. 5a) and the highest variability in POC/PON (CV = 38.7 %, Fig. 5d). Moreover, the slope of the best-fit function for CWD ($S$ = 8.35 ± 0.38) deviates significantly from the relationship corresponding to the canonical Redfield ratio. It is also notable that the intercept differs significantly from 0 ($I$ = -5.95 ± 2.20, Fig. 5a) which indicates an excess of PON relative to POC. The median of POC/PON for CWD is 7.3 (Table 1, Fig. 5d) which is significantly higher than the Redfield ratio and highest among the three data subsets. For the Arctic dataset (AOD), POC and PON are highly correlated ($R$ = 0.91, Fig. 5a). The variation in

POC/PON is large with the CV value of 30.9 % (Fig. 5c), which is twice as high as in the open-ocean dataset but somewhat lower than in the coastal-water dataset. The median of POC/PON for AOD is 6.9 which is closer to the Redfield ratio than the median values for OOD and CWD.





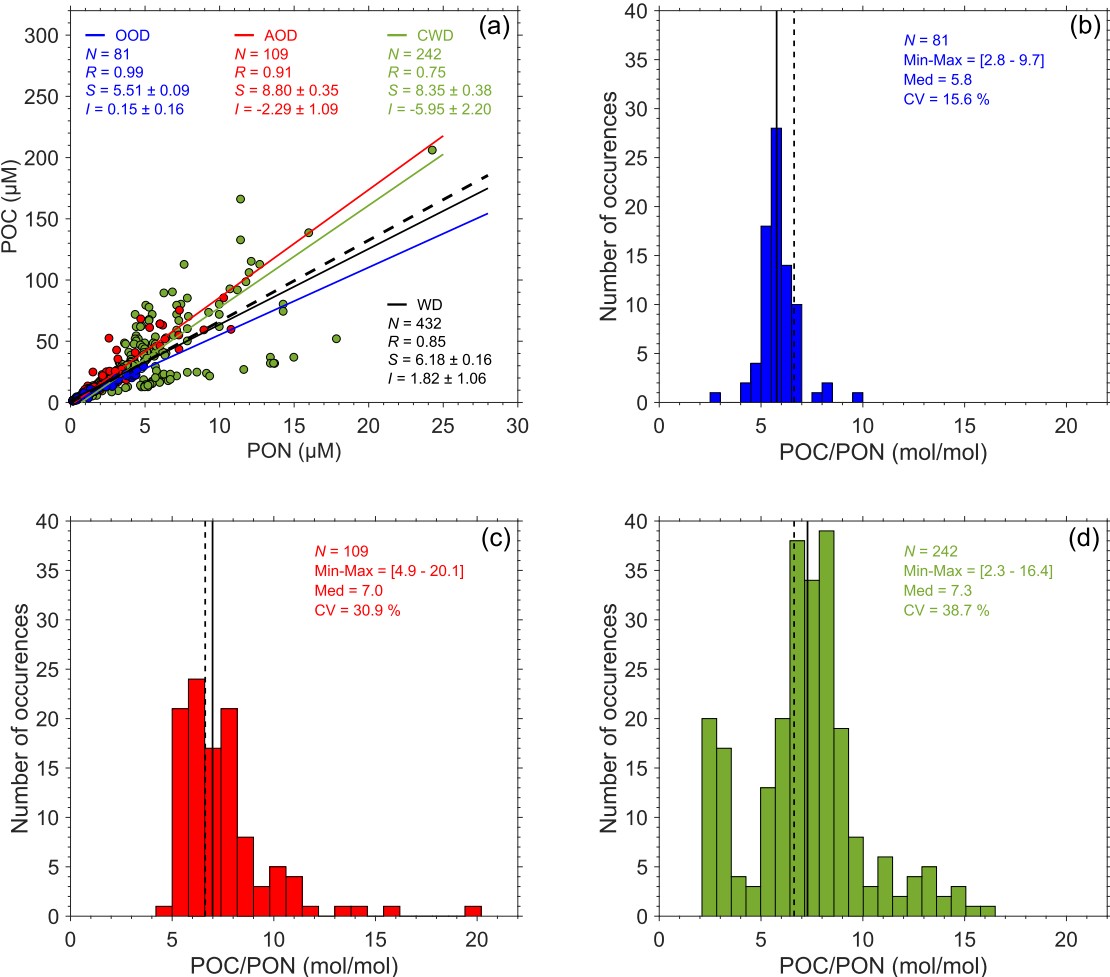

**Figure 5. (a) Scatter plot of PON as a function of POC where the data points are color coded to distinguish between the three data subsets OOD, AOD, and CWD. The blue, red, and green solid lines define the Model-II best-fit linear regression functions calculated for the OOD, AOD and CWD datasets, respectively. The black solid line denotes the best-fit function for the whole dataset WD. The black dashed line shows the Redfield ratio (POC/PON = 106/16 mol/mol). The number of data points, *N*, the coefficient of correlation, *R*, and the slope, *S*, and intercept *I* of the best-fit linear functions are indicated. The standard deviation of the slope, *S*, and intercept, *I*, is indicated. (b) Frequency distribution of the POC/PON ratio (mol/mol) for the open-ocean dataset OOW, (c) same as panel (b) but for the Arctic Ocean dataset AOD, and (d) same as panel (b) but for the coastal-water dataset CWD. In panels (b), (c), and (d) the vertical black solid lines correspond to the median values. The vertical black dashed line refers to the Redfield ratio value of 6.625. The values for the minimum-to-maximum range (Min-Max), median (Med), and coefficient of variation (CV) are also provided.**

The results presented in Fig. 5 are consistent with expectations regarding the variability in POC/PON in aquatic environments and previous reports on such variability. For example, Geider and Laroche (2002) compiled data on oceanic POC/PON from multiple sources and reported on a large range of variation between 3.4 and 12.5 mol/mol. It has also been known that the variation in inland waters is even larger with POC/PON reaching the values that are much higher than the





Redfield ratio (Bauer et al., 2013). The compilation of data from different inland aquatic environments showed the POC/PON range of 7.5−22.6 for lake environments (They et al., 2017) and 6.5−15.7 for rivers (Liu et al., 2020). Many measurements in our coastal-water dataset CWD were collected in areas affected by river plumes and associated input of terrestrial particulate organic matter which explains the large variability including the presence of high values of POC/PON in this dataset (Fig. 5a, 5d). The lowest values of this ratio (< 4) in CWD correspond to data collected in the northern Adriatic Sea and are consistent

with the previously reported values from this environment (Faganeli et al., 1989). The observed general trend of an increase in POC/PON variability from the open-ocean to coastal-water dataset reflects the increased complexity of the factors that drive the variation in the elemental composition of the bulk particulate organic matter. Different patterns of variability and deviations of measured POC/PON from the canonical Redfield ratio can be attributed to regional variations in environmental conditions, plankton biodiversity (Martiny et al., 2013), and carbon-enriched terrestrial inputs and/or preferential remineralization of PON

relative to POC (Dauby et al., 1994; Engel et al., 2001; Ferrari et al., 2003). Overall, the results of the PON/POC variability support the notion that PON cannot be reliably estimated from POC using the assumption of the Redfield ratio.

**3.2 Development of PON vs. IOP Relationships**

The particulate IOPs, $b_{bp}(\lambda)$, $a_p(\lambda)$, $a_{ph}(\lambda)$, and $a_d(\lambda)$, available in our dataset were measured at multiple light wavelengths as described in Sect. 2.2. For development of the relationships between PON and IOPs, our primary interest is in examining the

IOPs at selected light wavelengths that are consistent with the spectral bands used on several past and current satellite ocean color sensors. Figure 6 depicts the spectral pattern of the coefficient of determination, $R^2$, between $\log_{10}$-transformed PON and the four particulate IOPs at selected light wavelengths. In these calculations, the number of selected wavelengths is smaller for $b_{bp}(\lambda)$ (Fig. 6a) than for the absorption coefficients (Fig. 6b,c,d), which is associated with the spectral coverage of these measurements in our dataset. In addition, the results in Fig. 6 are shown for the whole dataset (WD) and separately for the

three data subsets, OOD, AOD, and CWD. In general, the spectral patterns of $R^2$ for most illustrated cases are relatively flat. A few exceptions include the spectral variations of $R^2$ for the PON vs. $a_d(\lambda)$ relationship, especially for the open-ocean dataset (Fig. 6d), as well as some variations associated with main spectral bands of phytoplankton absorption for the relationships involving $a_p(\lambda)$ and $a_{ph}(\lambda)$ (Fig. 6b,c). The $R^2$ values are generally substantially lower for the coastal-water dataset CWD compared to the OOD and AOD datasets. This result is expected given the largest variability in PON and IOPs in CWD.

In subsequent sections we present the relationships between PON and IOPs for a few selected wavelengths, specifically PON vs. $b_{bp}(\lambda)$ at 555 nm, and PON vs. $a_p(\lambda)$, $a_{ph}(\lambda)$, and $a_d(\lambda)$ at 442 nm and 510 nm. At these wavelengths, the $R^2$ values in Fig. 6 are either close to the maximum or remain relatively high within the spectral pattern of $R^2$. A more complete set of the relationships for other wavelengths that are commonly used on satellite ocean color sensors is provided in Supplementary Material (Tables S1 to S3).





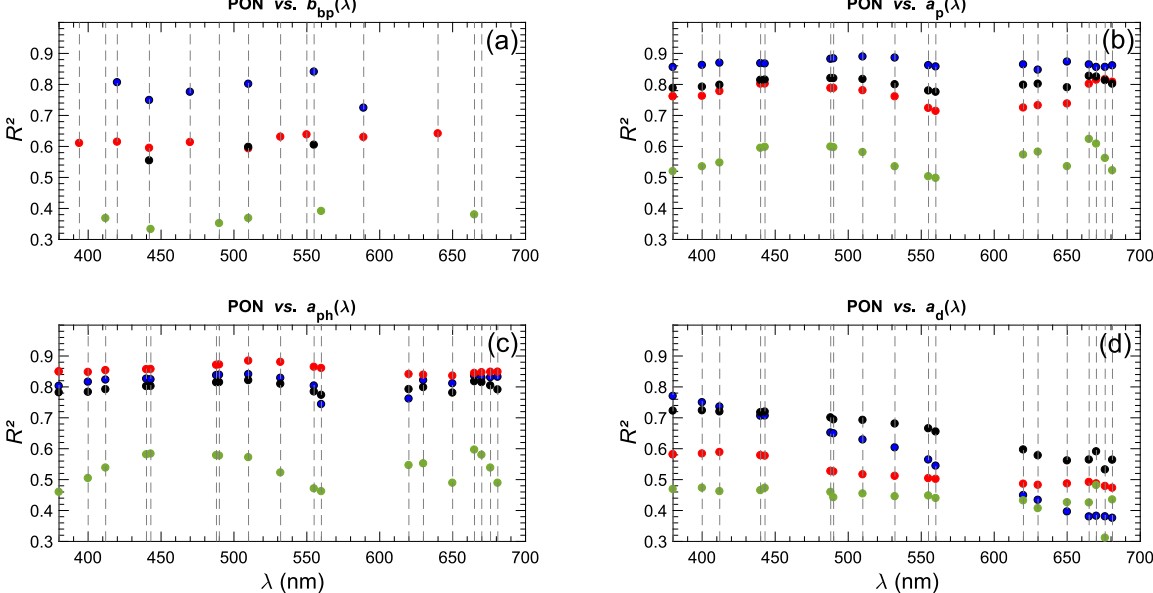


**Figure 6. Coefficient of determination ($R^2$) between log$_{10}$-transformed PON and particulate IOPs as a function of light wavelength. (a) PON vs. $b_{bp}(\lambda)$, (b) PON vs. $a_p(\lambda)$, (c) PON vs. $a_{ph}(\lambda)$, and (d) PON vs. $a_d(\lambda)$. The black, blue, red, and green circles refer to $R^2$ calculated at selected light wavelengths for the whole dataset (WD), the open-ocean (OOD), Arctic Ocean (AOD), and coastal-water (CWD) datasets, respectively. The data points for $R^2$ vs. $b_{bp}(\lambda)$ in panel (a) are at the following wavelengths: 394, 412, 420, 442, 470,**

**490, 510, 532, 550, 555, 560, 589, 640, 665 nm. The selected wavelengths for the absorption coefficients in panels (b), (c), and (d) are: 380, 400, 412, 440, 443, 488, 490, 510, 532, 555, 560, 620, 630, 650, 665, 670, 676, 689 nm.**

### 3.3 Relationship Between PON and Backscattering Coefficient

Figure 7a depicts the relationship between PON and $b_{bp}(555)$. When the open-ocean dataset (OOD) is considered, the scatter of data is relatively small (blue circles) and the pattern of data suggests that the relationship can be reasonably well described

by a power function (dark blue line):

$$\text{PON} = 105514.11 \, (\pm 72139.11) \, b_{bp}(555)^{1.31(\pm 0.07)} \tag{2}$$

where the values in parenthesis indicate the standard deviation of the best fit coefficients. For this subset of data, the coefficient

of determination between the log$_{10}$-transformed data is high ($R^2 = 0.84$). For comparison, Fig. 7a also includes a relationship recently established in the open-ocean waters of the western tropical South Pacific using in situ measurements of $b_{bp}$ from BGC-Argo floats (Fumenia et al., 2020). While the number of these data is comparatively small and the data cover a relatively narrow range of PON from about 0.28 to 13.3 mg m$^{-3}$, it is notable that this relationship is consistent with the relationship described by Eq. (2) for our larger OOD dataset.

When the Arctic Ocean (AOD) and coastal-water (CWD) datasets are considered, the scatter of data points is much larger and the relationships between PON and $b_{bp}(555)$ are considerably weaker. The $R^2$ values for these two datasets drop to 0.57



and 0.47, respectively. As a result, the relationship for the whole dataset (WD) is also relatively weak with a moderate $R^2$ of 0.63. Importantly, the overall pattern of all data in WD no longer suggests that a single power function of PON vs. $b_{bp}(555)$ can provide a reasonable description of the general trend of data observed across this whole dataset. In this case, the pattern of
data suggests that the relationship can be reasonably well described by a 3rd-degree polynomial function (black line):

$$PON = 10^{[2.62+1.32\ log_{10}(\ bbp(555))+0.68\ [log_{10}(\ bbp(555))]^2+0.14\ [log_{10}(\ bbp(555))]^3]} \quad (3)$$

Comparison of the algorithm-derived with measured values of PON is presented in Fig. 7b. This plot and the associated
statistical metrics provide a means to evaluate how the best-fit 3rd-degree polynomial function of PON vs. $b_{bp}(555)$ for the whole dataset WD from Fig. 7a reproduces the PON variability for this algorithm development dataset. Figure 7b shows a deviation between the linear fit to data and the 1:1 line which indicates that the $b_{bp}(555)$-based algorithm overestimates the PON values at low PON and tends to underestimate at high PON. Recognizing these biasing effects at different PON ranges is important, especially that *MdR* is very close to 1 indicating that an aggregate bias for the entire dataset WD is very small.
Other statistical indicators displayed in Fig. 7b are related to significant scatter of data points around the 1:1 line, for example *MdAPD* is about 35.9 %.

The results in Fig. 7a,b demonstrate that $b_{bp}(555)$ cannot be used as a good proxy of PON within a wide range of PON and $b_{bp}(555)$ variability observed across diverse marine bio-optical environments. This conclusion also holds for the backscattering coefficient at other light wavelengths (not shown), and is not surprising because the various physical-chemical
characteristics of natural particulate assemblages, which affect the optical properties of particles, are highly variable across diverse environments. Also, this conclusion is consistent with earlier studies that examined the estimation of POC from optical measurements, including the backscattering coefficient, across a wide range of aquatic environments. For example, a recent study of measurements from the western Arctic seas which exhibit a large range of variability demonstrated that the generally poor relationships based indiscriminately on all data can be improved by accounting for variations in the composition of
particulate matter parameterized in terms of POC/SPM ratio (Stramski et al., 2023). This is because this ratio can serve as a proxy of the contribution of organic particles to total suspended particulate matter which also includes mineral particles. In turn, the proportion of organic and mineral particles is one of important drivers of variations in particle optical properties, for example through changes in particle refractive index. Whereas other particle characteristics such as size distribution, shape, or degree of aggregation are also important determinants of particle optical properties, the changes in the compositional parameter
POC/SPM are useful in explaining, at least partly, the variability in the relationships between the measures of particulate organic concentration, such as PON and POC, and the bulk optical properties of seawater. Figure 7c depicts the variations in the PON-specific backscattering coefficient, $b_{bp}(555)$/PON, as a function of POC/SPM for our whole dataset. The variations span 2 orders of magnitude with a clear trend for a large decrease of PON-specific backscattering coefficient with an increase in POC/SPM ($R = 0.85$) which represents an increase in the proportion of organic particles in the suspended particulate matter.
For data corresponding to mineral-dominated particulate assemblages (POC/SPM < 0.12), the median value of $b_{bp}(555)$/PON





= $9.91 \times 10^{-4}$ m² mg⁻¹. The median drops by a factor of about 9 to a value of $1.07 \times 10^{-4}$ m² mg⁻¹ for data encompassing the organic-dominated assemblages (POC/SPM > 0.28). To first order, this trend is attributable to the fact that while organic particles contribute to both PON and $b_{bp}$, the mineral particles contribute only to $b_{bp}$. Overall, these results support the notion that, similar to the POC-specific particulate backscattering coefficient, the PON-specific particulate backscattering coefficient

is also strongly dependent on particulate composition.

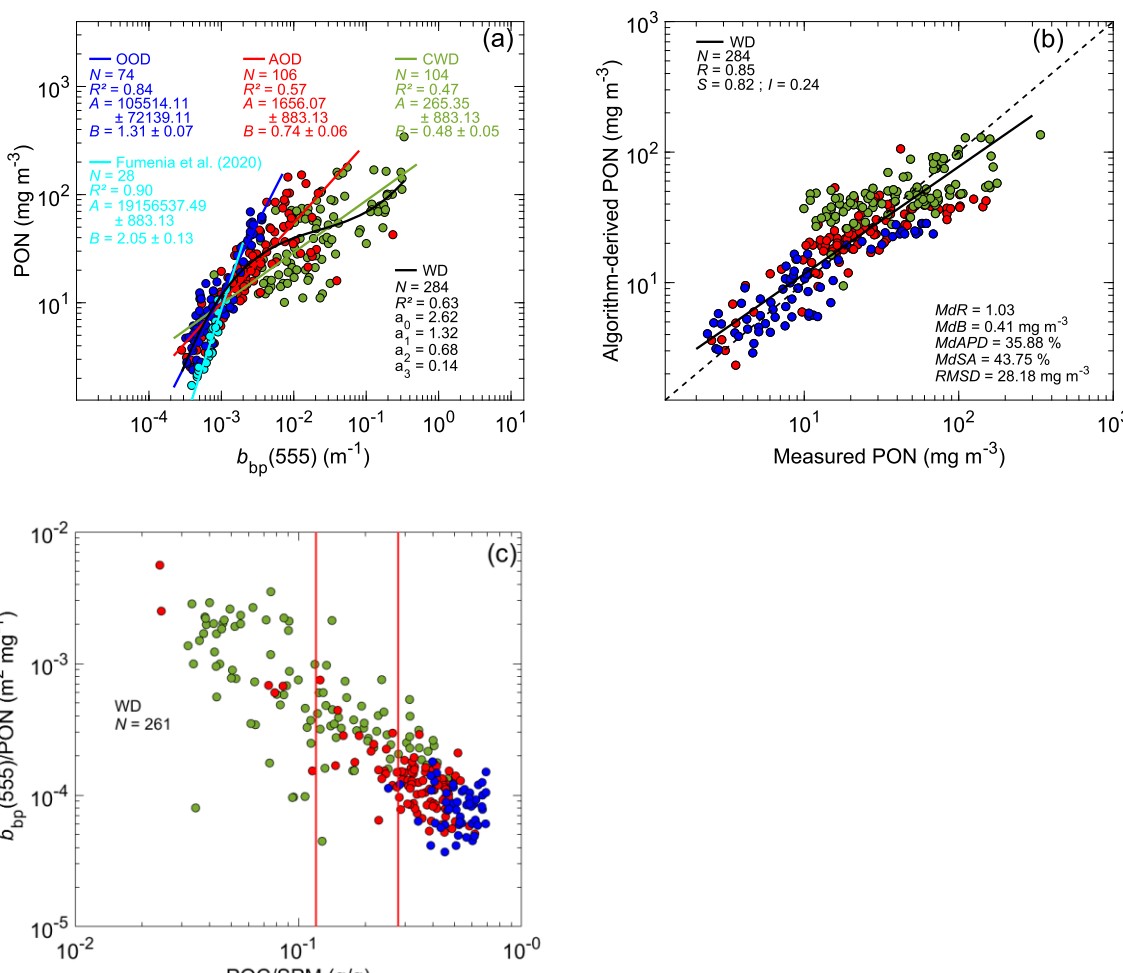

**Figure 7. (a) Relationships between PON and $b_{bp}$(555) for different datasets as indicated with black, dark blue, red, and green colors corresponding to the whole dataset (WD), open-ocean dataset (OOD), Arctic Ocean dataset (AOD) , and coastal-water dataset**
**(CWD), respectively. The solid dark blue, red, and green lines represent the best fit power functions obtained from Model-II linear regression on log₁₀-transformed data. The solid black line represents the best fit 3rd-degree polynomial function. For comparison, data in light blue from Fumenia et al. (2020) are shown after conversion of measured $b_{bp}$(700) to $b_{bp}$(555) using a $\lambda^{-1}$ spectral dependency. The standard deviation of the best fit coefficients, $A$ and $B$, is indicated. (b) Comparison of algorithm-derived and measured PON where the algorithm is the black line in panel (a) for the whole dataset WD. The black solid line is the best-fit function**
**obtained from Model-II linear regression on log₁₀-transformed data. The dashed line represents the 1:1 line. (c) Scatter plot of $b_{bp}$(555)/PON vs. POC/SPM. The red vertical lines refer to the threshold values of POC/SPM of 0.12 and 0.28 which delimit the mineral-dominated from mixed particulate assemblages and mixed from organic-dominated particulate assemblages, respectively (Stramski et al., 2023). Panels (a) and (b) of the figure include the statistical indicators (see Sect. 2.4 for details).**



### 3.4 Relationships Between PON and Absorption Coefficients

We now turn to relationships between PON and absorption coefficients, specifically the total particulate absorption coefficient, $a_p(\lambda)$, and its phytoplankton $a_{ph}(\lambda)$ and non-algal $a_d(\lambda)$ components. Figure 8a,b depicts data of PON vs. $a_p(\lambda)$ for two selected light wavelengths, 442 and 510 nm. In contrast to results for $b_{bp}(555)$ (Fig. 7a), the best-fit power functions for PON vs. $a_p(\lambda)$ are similar for the whole dataset WD and its subsets OOD, AOD, and CWD considered separately. This result indicates a relatively weak sensitivity of $a_p$-based PON algorithms to natural variability observed across diverse marine bio-optical

environments. For WD the determination coefficient $R^2 = 0.82$ which is much higher compared with 0.63 for the relationship based on $b_{bp}(555)$. Also, the scatter of all data points around the best-fit functions of PON vs. $a_p(442)$ or $a_p(510)$ is largely reduced compared to the relationship of PON vs. $b_{bp}(555)$. The best-fit power functions for the whole dataset WD are:

$$PON = 152.42 \ (\pm 9.50) \ a_p(442)^{0.65 \ (\pm 0.02)} \tag{4}$$
$$PON = 254.27 \ (\pm 19.62) \ a_p(510)^{0.64 \ (\pm 0.02)} \tag{5}$$

It is also notable that if the data subsets OOD, AOD, and CWD are considered separately, the relationships between PON vs. $a_p(\lambda)$ are strongest for the open-ocean dataset ($R^2 = 0.86$ or 0.89) and progressively weaken through the Arctic to the coastal-water dataset (for the latter $R^2 = 0.60$ or 0.58; Fig. 8a,b).

Figure 8c,d supports a reasonably good agreement between PON derived from the $a_p$-based algorithms and measured PON over the whole range of variability observed within the WD dataset. The best-fit regression functions of algorithm-derived vs. measured PON do not exhibit large deviations from the 1:1 line. The aggregate bias is negligibly small with $MdR = 1$ or 1.02. For all data in WD, $a_p(442)$ and $a_p(510)$ reproduce the PON variability with $MdAPD$ values slightly below 30.5 %, which is an improvement compared with 35.9 % for the PON estimation from $b_{bp}(555)$.






**Figure 8. (a) Relationships between PON and $a_p(442)$ for different datasets as indicated with black, dark blue, red, and green colors corresponding to the whole dataset (WD), open-ocean dataset (OOD), Arctic Ocean dataset (AOD), and coastal-water dataset**
**(CWD), respectively. The solid lines represent the best-fit power functions obtained from Model-II linear regression on log₁₀-transformed data. The standard deviation of the best fit coefficients, *A* and *B*, is indicated. (b) Same as panel (a) but for $a_p(510)$. (c) Comparison of algorithm-derived and measured PON where the $a_p(442)$-based algorithm is the black line in panel (a) for the whole dataset WD. The black solid line is the best-fit function obtained from Model-II linear regression on log₁₀-transformed data. The dashed line represents the 1:1 line. (d) Same as panel (c) but for the $a_p(510)$-based algorithm. (e) Scatter plot of $a_p(442)/$PON vs.**
**POC/SPM. The red vertical lines refer to the threshold values of POC/SPM of 0.12 and 0.28 which delimit the mineral-dominated from mixed particulate assemblages and mixed from organic-dominated particulate assemblages, respectively (Stramski et al., 2023). (f) Same as panel (e) but for $a_p(510)/$PON. Panels (a), (b), (c), and (d) of the figure include the statistical indicators (see Sect. 2.4 for details).**

Figure 8e,f shows that the PON-specific particulate absorption coefficients, $a_p(442)/$PON and $a_p(510)/$PON, exhibit

variations spanning more than 1 order of magnitude with a decreasing trend associated with an increase in POC/SPM ($R =$

0.66 and 0.63 for two selected light wavelengths, 442 and 510 nm, respectively). The trend is accompanied by about 3-fold

decrease in the median value of PON-specific $a_p(\lambda)$, for example the median of $a_p(510)/$PON decreases from $5.5 \times 10^{-2}$ m² mg⁻

¹ to $1.6 \times 10^{-2}$ m² mg⁻¹ as the composition of particulate matter changes from mineral dominated to organic dominated. While

the variations in PON-specific $a_p(\lambda)$ have impact on the ability to predict PON from $a_p(\lambda)$, these effects are weaker compared

to the case of particulate backscattering coefficient.

   Figure 9 depicts similar results to Fig. 8 but for the phytoplankton absorption coefficients, $a_{ph}(442)$ and $a_{ph}(510)$. Although

PON is associated with both phytoplankton and non-algal organic particles and the experimental determinations of $a_{ph}(\lambda)$ are

intended to represent the light absorption only by phytoplankton pigments, Fig. 9 shows that $a_{ph}(\lambda)$ has the ability to predict

PON which is comparable to the total particulate absorption coefficient $a_p(\lambda)$. The best-fit functions for the whole dataset WD

are (Fig. 9a,b):

PON = 220.73 ($\pm$ 16.96) $a_{ph}(442)^{0.67\,(\pm\,0.02)}$                                                                                           (6)

PON = 359.60 ($\pm$ 31.47) $a_{ph}(510)^{0.64\,(\pm\,0.02)}$                                                                                           (7)


   For these $a_{ph}$-based algorithms, the statistical metrics based on comparisons of algorithm-derived and measured PON (Fig.

9c,d) are very similar to those for $a_p$-based algorithms (Fig. 8c,d). For example, the *MdR* values remain very close to 1 and

*MdAPD* is slightly below 30 %. When separate subsets of data are considered, the relationship between PON and $a_{ph}(\lambda)$ is

strongest for the open-ocean and Arctic datasets and weaker for the coastal-water dataset. As expected, the PON-specific

phytoplankton absorption coefficient is weakly correlated with the proportion of organic particles in the suspended particulate

matter ($R = 0.47$ and 0.46 for two selected light wavelengths, 442 and 510 nm, respectively, figure not shown).





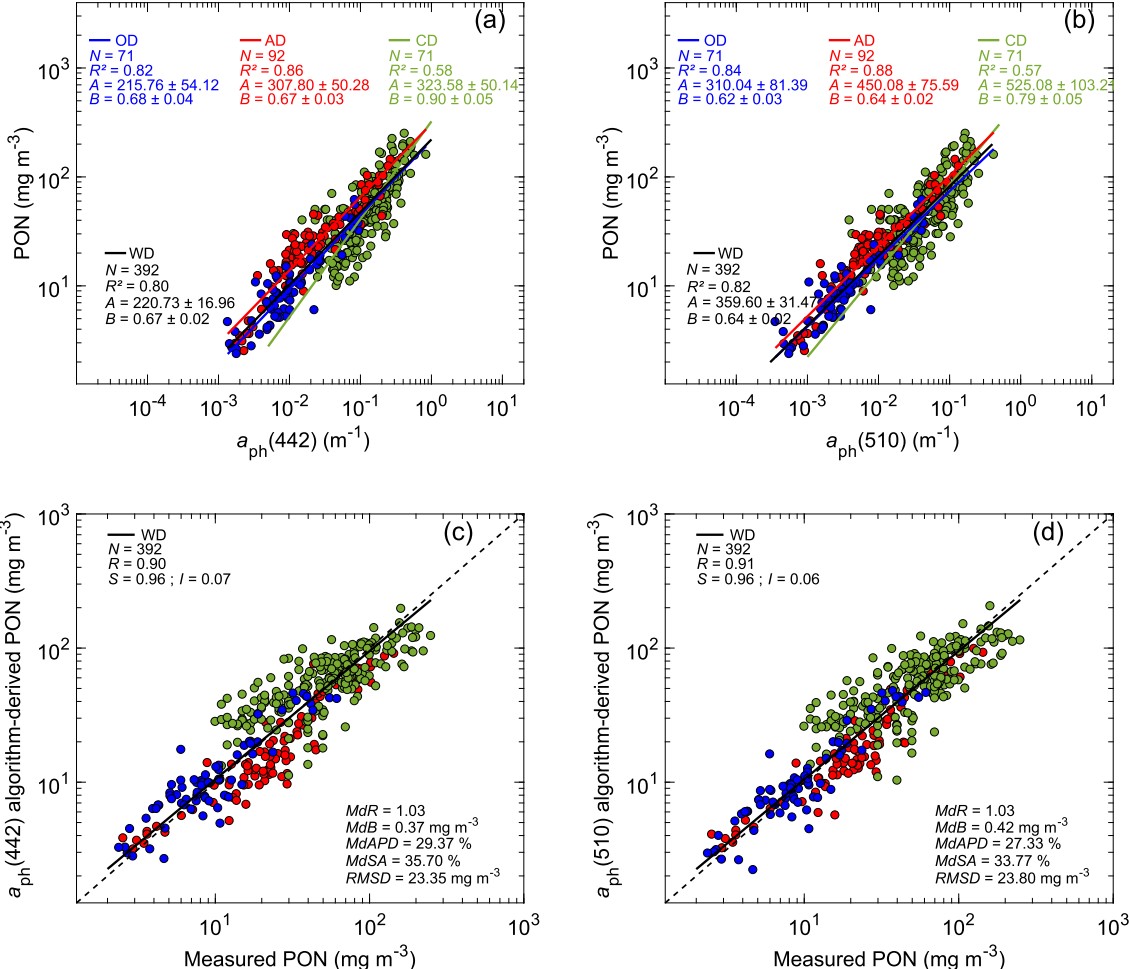

**Figure 9. (a) Relationships between PON and $a_{ph}(442)$ for different datasets as indicated with black, dark blue, red, and green colors**
**corresponding to the whole dataset (WD), open-ocean dataset (OOD), Arctic Ocean dataset (AOD), and coastal-water dataset (CWD), respectively. The solid lines represent the best-fit power functions obtained from Model-II linear regression on $\log_{10}$-transformed data. The standard deviation of the best fit coefficients, $A$ and $B$, is indicated. (b) Same as panel (a) but for $a_{ph}(510)$. (c) Comparison of algorithm-derived and measured PON where the $a_{ph}(442)$-based algorithm is the black line in panel (a) for the whole dataset WD. The black solid line is the best-fit function obtained from Model-II linear regression on $\log_{10}$-transformed data. The**
**dashed line represents the 1:1 line. (d) Same as panel (c) but for the $a_{ph}(510)$-based algorithm. Each panel of the figure includes the statistical indicators (see Sect. 2.4 for details).**




In contrast to $a_p$-based and $a_{ph}$-based PON algorithms, the PON vs. $a_d(\lambda)$ relationships are not as strong although they exhibit less inter-dataset variability compared to $b_{bp}$-based relationships (Fig. 10a,b). In particular, the slope of the best-fit functions

for the open-ocean dataset (OOD) differs significantly from the best-fit functions for the AOD and CWD datasets. The best-fit functions for the whole dataset WD are:

$$\text{PON} = 223.29 \, (\pm 21.18) \, a_d(442)^{0.55 \, (\pm 0.02)} \tag{8}$$

$$\text{PON} = 351.94 \, (\pm 42.41) \, a_d(510)^{0.57 \, (\pm 0.02)} \tag{9}$$


but these $a_d$-based algorithms are inferior to $a_p$-based and $a_{ph}$-based algorithms for estimating PON across a wide dynamic range of PON and IOPs observed within the whole dataset. Figure 10c,d shows increased uncertainty of PON predicted from Eqs.(7) and (8) for all data included in WD, which is manifested especially through the increased values of *MdB*, *MdAPD*, *MdSA*, and *RMSD*. Variations in the particulate composition, as parameterized by POC/SPM ratio, exert a similar influence

on the $a_d(\lambda)$/PON ratio ($R = 0.67$ and $0.61$, for two selected light wavelengths, 442 and 510 nm, respectively, Fig. 10e,f) to the case of the $a_p(\lambda)$/PON ratio (Fig. 8e,f).







**Figure 10. Same as Figure 8 but for the non-algal particulate absorption coefficients $a_d$(442) and $a_d$(510).**



## 5 Concluding Remarks

The analysis of empirical relationships between PON and particulate IOPs indicate that the total particulate, $a_p(\lambda)$, and phytoplankton, $a_{ph}(\lambda)$, absorption coefficients have potential to be used as predictive variables in IOP-based algorithms for estimating PON over a wide range of oceanic environments. Specifically, the analysis of near-surface measurements from various open ocean and coastal regions including Arctic waters show consistent relationships of PON vs. $a_p(\lambda)$ or vs. $a_{ph}(\lambda)$ across datasets collected in different marine bio-optical environments. For the whole dataset considered in this study to formulate the $a_p$-based and $a_{ph}$-based algorithms in the form of power functions, a median absolute percent difference between the algorithm-derived and measured PON is slightly less than 30 %. The relationships of PON vs. non-algal particulate absorption coefficient, $a_d(\lambda)$, or vs. particulate backscattering coefficient, $b_{bp}(\lambda)$, are weaker, especially when a wide dynamic range of PON and IOPs within the whole dataset is considered. However, for the subset of data from open ocean waters, our results indicate that $b_{bp}(555)$ can serve as a reasonably good proxy of PON and this result is consistent with earlier data from a geographically more restricted region of the western tropical Pacific (Fumenia et al., 2020).

To further support these conclusions, a comparative assessment of the goodness-of-fit associated with the different versions of IOP-based algorithms when evaluated with the whole dataset (WD) as well as its component subsets, open-ocean dataset (OOD), Arctic Ocean dataset (AOD), and coastal-water dataset (CWD) is presented in Fig. 11. This figure illustrates the areas of polygons created by six statistical indicators which include $R$, $S$, $MdB$, $MdAPD$, $MdSA$, and $RMSD$ after appropriate normalization (Sect. 2.4). The size of the polygon area is related to quality of goodness-of-fit for a given algorithm as evaluated with its development dataset. As the area of polygon becomes smaller, the overall representation of PON variability by the algorithm improves, and hence the algorithm has greater potential for better performance. Figure 11 supports the conclusion that the $a_p$-based and $a_{ph}$-based algorithms best represent the PON variability over a large dynamic range within the whole dataset WD as well as within separate data subsets OOD, AOD, and CWD. The $b_{bp}$-based algorithm may perform reasonably well only in open ocean waters. This result can be relevant to efforts aiming at estimation of PON from optical sensors deployed on in situ autonomous platforms such as BGC-Argo floats.

Given the relative scarcity of concurrent PON and IOP measurements across a wide range of marine bio-optical environments, in the present study all available data were used to examine and formulate the PON vs. IOP relationships and no independent data were available for validation. In general, the performance of the algorithms is limited by the variability in the relationships between PON and particulate IOPs which, in turn, is associated with multiple factors, such as variations in the composition and size distribution of suspended particulate matter, which drive variations in PON-specific IOP coefficients. In this study we demonstrate these effects by showing the variations in PON-specific IOP coefficients with changes in POC/SPM ratio that provides information about relative proportions of organic and mineral particles. Accounting for such variability appears desirable if further improvements are to be achieved in the performance of optically-based PON algorithms across diverse oceanic environments. This research avenue has been recently described in relation to POC algorithms (Stramski et al., 2023; Koestner et al., 2024). To further support this research, there is a clear need to collect more concurrent



measurements in diverse aquatic environments of seawater optical properties and various characteristics of suspended particles
including the measures of particulate concentration such as PON, POC, and SPM as well as some measures of particle size
and composition which also play important roles in bio-optical variability. The availability of more extensive datasets can
serve the purpose of both the improved algorithm development and validation with independent data.

**Figure 11. Radar plots summarizing the performance of the IOP-based algorithms for deriving PON. The IOPs(λ) considered are
the $b_{bp}(555)$ (blue line), $a_p(442)$ (red line), $a_{ph}(442)$ (yellow line), and $a_d(442)$ (purple line). The smallest area of the polygon associated
with each algorithm represented in the radar plot corresponds to the best performance. The coefficient of correlation, *R*, Slope, *S*,
Root Mean Square Deviation, *RMSD*, Median Bias, *MdB*, Median Absolute Percentage Difference, *MdAPD*, and Median Symmetric
Accuracy, *MdSA*, subject to appropriate normalization (see Sect. 2.4) are indicated (a) for the WD, (b) OOD, (c) AOD, and (d) CWD
datasets.**

**Data availability**

The majority of data used in this study are publicly available from the following online data repositories: the LEFE-CYBER database (http://www.obs-vlfr.fr/proof/index_vt.htm; BIOSOPE), the NASA SeaWiFS Bio-optical Archive and Storage System (https://seabass.gsfc.nasa.gov/; HLY1001, HLY1101), the PANGAEA Data Publisher for Earth and Environmental
Science (https://doi.org/10.1594/PANGAEA.902230; ANTXXVI/4, KM12-10), the Data and Sample Research System for Whole Cruise Information database of the Japan Agency for Marine-Earth Sciences (https://doi.org/10.17596/0001879; MR1705-C), and the SEA scieNtific Open data Edition (https://www.seanoe.org/data/00824/93570/; COASTlOOC).

**Author contributions**

AF, HL, RR, and DS conceptualized the study. HL contributed to the methodology and the supervision of this study. RR
performed the data curation. DS and RR contributed to the resources. AF developed the computer code and supporting algorithms and performed the visualization. AF, HL and DS contributed to the writing of the original draft preparation. AF, RR and DS contributed to the investigation. AF, HL, and DS contributed to the acquisition of funding. All authors contributed to the writing review and editing.

**Competing interests**

The authors declare that they have no conflict of interest.

**Acknowledgements**

This work was supported by Centre National d'Etudes Spatiales in the frame of the COUP-PNP project (CNES/ TOSCA program), the postdoc funding of Alain Fumenia by the National Center for Space studies (CNES), the ANR CO2COAST project (ANR-20-CE01-0021) awarded to H. Loisel, and by the NASA PACE project (80NSSC20M0252 awarded to D.
Stramski and R. A. Reynolds). Portions of this work were performed during the stay of Hubert Loisel at Scripps Institution of Oceanography in 2023. The investigated datasets were assembled from in situ measurements made on different cruises and field experiments. We thank all scientists and crew involved in fieldwork for their support and contributions to collection of data.





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
