# Peer review of "Relationships between the concentration of particulate organic nitrogen and the inherent optical properties of seawater in oceanic surface waters"

_EGUsphere, 2024_

## Author Comment (AC2)

**Responses to Referee #2**

"Relationships between the concentration of particulate organic nitrogen and the inherent optical properties of seawater in oceanic surface waters"

Author(s): Alain Fumenia, Hubert Loisel, Rick A. Reynolds and Dariusz Stramski

*General Comments by Referee*

*This paper presents a comprehensive analysis of relationships between surface ocean PON and various IOPs and proposes various algorithms for the estimation of PON from IOPs obtainable from ocean colour remote sensing or more directly from in situ measurements. This paper exploits -to the best of my knowledge- the largest available dataset of concurrent IOP and PON measurements and covers open ocean, Arctic and coastal environments. The paper is very well written and logically structured. All aspect of this paper, from the writing, to the statistical analyses and discussion are very rigorous and complete; I have nothing to add to this work. I applaud the authors for their rigorous and comprehensive work on the estimation of PON in seawater and I look forward to follow-up work!*

**Response**: We thank Dr. Griet Neukermans for her review and positive feedback on our manuscript. We greatly appreciate your positive assessment of different aspects of our study and the quality of presentation of this work in our manuscript. This is highly encouraging and motivating to continue the pursuit of this line of research.

---

## Author Response (AR1)

**Responses to Referee #1**

"Relationships between the concentration of particulate organic nitrogen and the inherent optical properties of seawater in oceanic surface waters"

Author(s): Alain Fumenia, Hubert Loisel, Rick A. Reynolds and Dariusz Stramski

We appreciate the constructive comments by the Referee. Here we provide our detailed point-by-point responses and a description of any actions taken in regard to these comments. The Referee's comments are shown in italicized font; our responses follow each comment in normal font. Line numbers and figures indicated in our responses refer to the revised manuscript unless otherwise noted.

*General Comments*
*The paper seeks to address the important issue of determining the Particulate Organic Nitrogen concentration of the surface ocean by using IOPs. IOPs are readily derivable from satellite ocean colour products and therefore (theoretically) global coverage can be achieved.*

*Although this paper is highly detailed, I found it to be an excellent and logical read, as the authors took the reader through the various different algorithmic approaches to deriving PON from spectral IOPs. It is very well written and the level of rigor in terms of looking at the different IOP types (bbp, then ap, then aph) as well as conclusively and quantitatively showing that using a Redfield stoichiometric approach only works in the open ocean, was convincing.*

*The next step I suppose for this paper would be actual application to broadscale satellite data; if such an application were made in this paper, it could have been bogged down in the strengths and weaknesses of the different algorithms (e.g. Lee etc).*

**Response:** We thank the Referee for positive and detailed feedback on our manuscript. We appreciate your perspective on the next steps for this research, particularly the application of the proposed algorithms to broad-scale satellite data. We agree that this application represents an important avenue for future work and plan to explore it in subsequent studies. We envision, for example, that the proposed PON algorithms based on IOPs can be applied to IOPs derived from remote-sensing reflectance, $R_{rs}(\lambda)$, using inverse optical models (e.g., Jorge et al., 2021; Kehrli et al., 2024; Loisel et al., 2018; Zheng and Stramski, 2013). A short section on potential applications to satellite data has been added to the concluding remarks in the revised manuscript (Page 30, lines 646-649).

*Although the authors talk about the biogeochemical importance of PON, little reference is made to this in the rest of the paper. A small section in the concluding remarks could be dedicated to this. What are the biogeochemical characteristics of water masses which are above / below Redfield for example, likely to be? What are the consequences for this on nutrient cycling, biological growth etc. These are small points, and likely conjecture on my part.*

**Response:** Your suggestion to include a section in the concluding remarks on the biogeochemical importance of PON is well-taken. We added a concise section to address this point (Page 30, lines 649-658).

*Specific Comments*

*P 9: Figure 2: It would be useful to have the error bars plotted (if available), although can be plotted based on the expected uncertainties.*

**Response:** We agree that error bars can enhance the interpretation of results. We have added the error bars in Figure 2 using the expected uncertainties derived from Loisel et al. (2001).

*Also not quite sure why the x- and y-axes are switched on a) and b). It would make most sense if the control variable was SPM in both cases.*

**Response:** We agree that using SPM as the control variable in Figures 2a and 2b could be a good option from the standpoint of consistency. However, we would like to clarify the reasoning behind the current presentation. The main goal of Figure 2a is to assess the reliability of LS-derived $b_{bp}(\lambda)$ by comparing it with $b_{bp}(555)$ obtained from an existing empirical relationship ($b_{bp}(555)$ vs. SPM) described by Neukermans et al. (2012). In this context, SPM is used as the control variable for predicting $b_{bp}$ from SPM. In contrast, Figure 2b serves as an inter-comparison illustration which includes the SPM vs. $b_{bp}(555)$ relationship developed by Stramski et al. (2023). Here, $b_{bp}(555)$ is used as the control variable for predicting SPM. While we understand the point associated with a preference for consistency, the differences in the formulation of original empirical relationships presented in Figures 2a and 2b explain the use of *x* and *y* variables.

*P30: Figure 11 – I am not particularly familiar with these type of plots, and I am not sure what they add overall to the findings of the paper.*

**Response:** Although the use of this type of visualization has not been very common, similar radar plots have been shown to be useful for comparison of algorithm performance (e.g., Bonelli et al., 2021; 2022; Tran et al., 2019). We believe that Figure 11 in our manuscript provides a useful synthetical illustration of comparative assessment of the goodness of fit parameters associated with the different IOP-based algorithms. To address the Referee's point, we have made some additions including new references to better clarify the purpose and interpretation of this figure (Page 29, lines 621-623).

*Technical Corrections*

*Figure 10 top panels have half of the x-axis label covered up*

**Response:** Thank you for pointing out this problem. The correction has been made.